# Laponite-Based Nanocomposite Hydrogels for Drug Delivery Applications

**DOI:** 10.3390/ph16060821

**Published:** 2023-05-31

**Authors:** Samuel T. Stealey, Akhilesh K. Gaharwar, Silviya Petrova Zustiak

**Affiliations:** 1Department of Biomedical Engineering, Saint Louis University, Saint Louis, MO 63103, USA; samuel.stealey@slu.edu; 2Department of Biomedical Engineering, Texas A&M University, College Station, TX 77433, USA; gaharwar@tamu.edu

**Keywords:** nanosilicate, nanoclay, nanocomposite, hydrogel, Laponite, drug delivery

## Abstract

Hydrogels are widely used for therapeutic delivery applications due to their biocompatibility, biodegradability, and ability to control release kinetics by tuning swelling and mechanical properties. However, their clinical utility is hampered by unfavorable pharmacokinetic properties, including high initial burst release and difficulty in achieving prolonged release, especially for small molecules (<500 Da). The incorporation of nanomaterials within hydrogels has emerged as viable option as a method to trap therapeutics within the hydrogel and sustain release kinetics. Specifically, two-dimensional nanosilicate particles offer a plethora of beneficial characteristics, including dually charged surfaces, degradability, and enhanced mechanical properties within hydrogels. The nanosilicate–hydrogel composite system offers benefits not obtainable by just one component, highlighting the need for detail characterization of these nanocomposite hydrogels. This review focuses on Laponite, a disc-shaped nanosilicate with diameter of 30 nm and thickness of 1 nm. The benefits of using Laponite within hydrogels are explored, as well as examples of Laponite–hydrogel composites currently being investigated for their ability to prolong the release of small molecules and macromolecules such as proteins. Future work will further characterize the interplay between nanosilicates, hydrogel polymer, and encapsulated therapeutics, and how each of these components affect release kinetics and mechanical properties.

## 1. Introduction

Hydrogels are widely used for drug delivery applications due to their potential for localized delivery of a variety of therapeutics while preserving drug bioactivity. However, a notable challenge with hydrogels is their susceptibility to the initial burst release of loaded therapeutics [1]. This is mostly attributed to a highly porous and hydrated network, which facilitates the diffusion of encapsulated drug molecules. To address this issue, several techniques have been employed to minimize this burst release and achieve sustained drug release kinetics [2]. These techniques include the utilization of stimuli-responsive polymers, the attachment of drugs to the polymer network, and the incorporation of nanomaterials into the polymeric network. 

The use of nanoparticles in reinforcing hydrogels and enhancing drug delivery is attractive due to their ability to improve mechanical properties, increase drug loading capacity, enable controlled and sustained drug release, and facilitate targeted delivery. Extensive research is being conducted on a wide range of nanoparticles, including polymeric, carbon-based, metal, metal oxides, and ceramic nanoparticles, to reinforce hydrogel networks for drug delivery applications [3]. Among the various nanomaterials being explored for nanocomposite hydrogel drug delivery devices, Laponite (nanosilicates) stands out as a particularly promising and emerging nanomaterial [4]. Laponite is a synthetic two-dimensional (2D) nanosilicate particle that has garnered significant attention in the field [5]. Laponite exhibits a unique shape and surface charge that makes it well-suited for drug delivery applications. Its structure consists of disc-shaped particles with a high aspect ratio, allowing for the efficient loading of therapeutic molecules. Due to discotic charge characteristics, Laponite also enhances interactions with cationic, anionic, or neutral molecules [6]. The combination of high surface area and charge also results in the sustained release of loaded therapeutics [7]. Interestingly, a range of therapeutics, including small molecule drugs, peptides, and large proteins, can be easily loaded and delivered using Laponite. 

As Laponite is highly hydrophilic, it can easily interact with a range of polymeric hydrogels. The addition of Laponite to the polymeric network has been shown to improve shear-thinning characteristics, which is important for the minimally invasive and localized delivery of therapeutics [8]. Moreover, Laponite addition has also been shown to improve the mechanical strength as well as physiological stability of polymeric hydrogels. A range of studies have demonstrated the high biocompatibility of Laponite, establishing its widespread application in biomedical applications. More recently, Laponite-loaded polymeric hydrogels received 510k approval from the US Food and Drug Administration (FDA) establishing their clinical potential [9]. 

In this review, we critically evaluate the use of Laponite-based nanocomposite hydrogels for drug delivery applications. Specifically, we focus on unique characteristics of Laponite, such as the shape, size, and charge, that make it attractive for drug delivery applications. Then, we examine the interactions of Laponite with various polymers to design nanocomposite hydrogels and explore its unique and beneficial properties. The ability of Laponite-based nanocomposite hydrogels for sustained and controlled release of small molecules and macromolecules such as proteins, is examined. In addition, we discuss the structure and stoichiometry of Laponite–therapeutics complexes, which can represent the next frontier in harnessing the utility of Laponite–hydrogel nanocomposites for drug delivery applications. 

## 2. Hydrogels in Drug Delivery Applications

Hydrogels are water-swollen polymer networks that have been extensively characterized and widely utilized for drug delivery applications [2]. Hydrogels can be fabricated from a wide variety of natural polymers (gelatin, collagen, hyaluronic acid, etc.) and/or synthetic polymers (polylactic acid, polyglycolic acid, polyethylene glycol, etc.). Hydrogels are attractive for drug delivery applications due to their highly biocompatible, ability to mimic the physical properties of most soft tissues, and capability to maintain the bioactivity of encapsulated therapeutics [10,11].

Furthermore, the mesh size and degradation profiles of hydrogels can be tunable based on polymer structure, molecular weight, crosslinking mechanism, and degradability [12]. For example, some hydrogels may undergo hydrolytic or enzymatic cleaving, leading to the release of entrapped therapeutics [13]. Slow-degrading hydrogels can release encapsulated cargo and then be degraded or resorbed, eliminating the need for device removal following release [14]. In most hydrogels, the release is controlled by diffusion, which is governed by the relative size ratio of the solute and the mesh size of hydrogels [15]. Small-molecule therapeutics (<500 Da) are typically rapidly released from hydrogels, where hydrogel mesh size is typically much larger than the small molecules [2]. Conversely, larger molecules of therapeutics (>10 kDa), such as proteins, will diffuse slowly [16]. This diffusion-controlled release by hydrogels can lead to high initial burst release, where a significant amount of the encapsulated cargo is rapidly release before achieving a stable sustained release profile [17,18]. While some applications such as wound healing may desire initial burst dosing, in some cases such a burst release can lead to undesirable pharmacokinetic properties [19,20]. A high concentration of the drug may lead to toxicity, while a low plasma concentration of the drug will prevent the desired therapeutic effect [21]. Undesired burst release can be economically wasteful, as significant quantities of drugs are rapidly cleared by the body [17]. 

It is desirable to design hydrogels with controlled and sustained release strategies that have been implemented to prolong the release of encapsulated therapeutics from hydrogels. For example, encapsulated therapeutics may be tethered to the polymer matrix, preventing rapid diffusion until linkages are cleaved [22,23]. Cleaving of these tethers offers good tunability, but may alter the structure of the drug, reducing its activity [24]. Stimuli-responsive hydrogels may have variable swelling properties based on local environmental pH, salinity, or temperature, thereby altering the release rate [25,26,27,28]. Drug release kinetics are, therefore, altered based on the swollen hydrogel mesh size, in which higher swelling leads to faster release. While stimuli-responsive hydrogels offer the potential for delayed and controlled release, swelling may be difficult to predict in a translational setting and may vary from patient to patient, thereby causing variance in release rates [29]. In another strategy, the incorporation of nanomaterials into hydrogels allows for physical or chemical entrapment of encapsulated cargo to allow for sustained release, which is the focus of this review. 

## 3. Nanocomposite Hydrogels in Drug Delivery

Nanomaterials, which are defined as materials that have at least one dimension in the range of 1–100 nm, exist in a wide variety of shapes and compositions, thereby leading to a range of interactions with therapeutic molecules [30]. For example, gold and silver nanoparticles have been widely explored as tethering agents [31,32,33]. These nanoparticles offer benefits, such as high drug loading, prolonged drug stability, and targeted delivery [34]. Therapeutics have also been entrapped within liposomes, preventing drug diffusion out of the hydrogel until the dissolution of the liposomal structure [35,36]. Mesoporous silica nanoparticles have been entrapped within hydrogels to provide a tortuous path for encapsulated cargo to diffuse, leading to sustained release [37,38]. Two-dimensional (2D) charged nanosilicates have also recently been incorporated into hydrogels to electrostatically adsorb drugs and prolong release kinetics [39,40,41] and are the main focus of this review. 

The incorporation of nanomaterials into hydrogel structures offers a variety of benefits, including augmented hydrogel physical and mechanical properties, adsorption/intercalation of drugs to mitigate burst release, physical crosslinking of polymers, and localization of therapeutic release. The nanocomposite hydrogels offer advantages not afforded by the individual components by themselves. For example, hydrogels may slow drug release, but are susceptible to diffusion-controlled burst release and difficulty in achieving sustained release. On the other hand, nanomaterials by themselves can offer sustained release, but these particles may be rapidly cleared, which is undesirable for localized delivery applications.

## 4. Two-Dimensional (2D) Nanosilicates

Silicate minerals, which are the largest group of minerals and consist of subunits with the formula [SiO_2+n_]^2n−^ balanced by metallic anionics, exist in a variety of naturally occurring and synthetic structures that can be utilized in a plethora of biomedical applications. Major types of silicates include nesosilicates, cyclosilicates, sorosilicates, inosilicates, tectosilicates, and phyllosilicates. These phyllosilicates typically form sheet structures made of hydrated aluminosilicates in a Si_2_O_5_ ratio [42]. Cations such as Mg, K, Na, Ca, and Fe may be naturally substituted into the structures, giving rise to desirable charge characteristics that may be used to interact with drugs for delivery applications. Phyllosilicate sheets are formed by stacked tetrahedral and octahedral sheets in a 1:1 or 2:1 ratio. The tetrahedral layers are formed by Si cations coordinated to O atoms in a hexagonal pattern, while the octahedral layer is formed by the coordination of metal cations with O, OH^−^, or F^−^ of the tetrahedral layer [43]. Phyllosilicates, also known as nanoclays, may exist in a layered or tubular structure. 

In 2:1 tetrahedral:octahedral nanoclays, the cationic substitution of aluminum is possible, leading to nanoclays containing magnesium, iron, lithium, and iron. These substitutions lead to a charge unbalance, giving rise to nanoclays with a varying net charge, surface reactivity, cationic exchange capacity, and swelling behaviors [44]. Nanoclays without cationic aluminum substitution (known as prophyllites) exhibit similar properties to 1:1 nanoclays, with low reactivity and low swelling behavior, making them less appropriate for delivery applications. Other 2:1 clay minerals are talc, illites, smectites, chlorites, and vermiculites. Illites and chlorites have low cationic exchange capacity and water retention, limiting their utility for drug delivery [45,46]. In a study by Lima et al., talc was loaded into chitosan-based hydrogels and exhibited high loading capacities and prolonged release of the anti-diuretic amiloride [47]. Interestingly, X-ray diffractograms revealed mostly surface adsorption of the drug as opposed to intercalation. Vermiculite has also been used in hydrogel composite delivery devices to deliver antibacterial compounds due to its relatively high cation exchange capacity [48,49]. 

Of these 2:1 nanoclays, smectites are the class with high swelling potential, making them the most investigated phyllosilicates for biomedical applications [50]. The various groups of smectites are listed in Table 1. The swelling behaviors of smectites make them attractive for biomedical applications, allowing for electrostatic adsorption of molecules as well as intercalation of molecules into the interlayer space, which is typically made of sodium or calcium ions and water molecules [51,52].

Of the smectites, montmorillonite is by far the most well-studied nanoclay for delivery applications due to its abundance in nature and high cationic exchange capacity. Montmorillonite has been incorporated into numerous hydrogel systems for the delivery of both small molecules and macromolecules [44,60,61,62,63]. Hectorite was used in an alginate hydrogel system developed by Joshi et al. to deliver quinine, a small-molecule antimalarial drug [64]. Similarly, saponite was also used to adsorb and deliver quinine by Kumeresan et al. [65]. Nontronite hydrogel composites have been shown to have variable swelling behavior based on local pH and salinity, offering the potential for release applications [66], but no delivery studies were found in the literature. Chitosan–beidellite composites were fabricated by Cheikh et al. and exhibited sustained release of diclofenac sodium, a model drug [67]. Recently, a burst of interest has been dedicated to studying Laponite for biomedical applications, including drug delivery. 

## 5. Laponite

Laponite is a synthetic hectorite with an octahedral layer consisting of magnesium and lithium cations with a diameter of ~30 nm and a layer thickness of ~1 nm. This high aspect ratio and surface area lends itself to a variety of biomedical applications, as Laponite is known to impart beneficial mechanical properties to hydrogels, such as stiffness and shear-thinning behavior (Figure 1) [68]. Laponite has a dual-charged nature, where particle edges are positively charged due to the charge imbalance caused by magnesium and/or lithium of octahedral aluminum, while the faces (top and bottom surfaces) are negatively charged as a result of the silicate tetrahedral layers. As such, both negatively charged and positively charged molecules may be electrostatically adsorbed to the surface of Laponite particles [7]. Neutral molecules may still interact with Laponite particles if their charge is anisotropically distributed [69]. Additionally, the high swelling behavior of Laponite particles allows for interlayer intercalation of molecules [70]. Thus, the charged nature of Laponite easily lends itself to drug delivery applications in which drug molecules may electrostatically interact with Laponite particles to slow diffusion and subsequent release. 

Laponite has been used for a variety of other applications as well. It has been shown to impart osteogenic and angiogenic potential in the absence of therapeutics [71,72]. The shear-thinning behavior of Laponite makes it attractive for bioprinting applications, as well as injectable biomaterials [4,73,74]. Outside of biomedical applications, Laponite has been used for wastewater treatment and a variety of industrial applications, such as a rheological modifier in cosmetics, cleaning products, and polymer films [75,76,77,78]. 

The synthesis of Laponite and synthetic hectorites has been well established, with a common procedure of crystallizing an aqueous mixture of LiF, Mg(OH)_2_, and SiO_2_ at high temperature [79,80]. Modification of the reactant molar ratios, heating method, and temperature can affect final product purity and size [81]. The review by Zhang et al. provides a deeper dive into how different synthetic hectorite fabrication methods can produce a variety of closely related structures. However, Laponite is typically produced and used commercially, with the brand name Laponite first introduced by Laporte Industries (now BYK) in the early 1960s [82]. 

## 6. Degradation and Cytotoxicity of Laponite

Laponite particles naturally dissociate into their constituent ions (Li^+^, Mg^+^, and Si(OH)) in environments where the local pH is less than that of the isoelectric point of Laponite (pH ~10) [83]. It has been hypothesized that in these lower pH environments, H^+^ ions react with the nanoclay and leach Mg^+^ and Li^+^ ions, thereby degrading the nanosilicate particles in about 20–50 days [55,84]. In vivo, Laponite particles are thought to be internalized by clathrin-mediated endocytosis and subsequently degraded within the low pH environment of endosomes (Figure 2) [83,85,86].

When used for therapeutic delivery applications, the delivery vehicle itself must not induce significant adverse effects. Gaharwar et al. demonstrated individual Laponite particles are non-cytotoxic beneath a critical concentration, with an IC50 of 4 mg/mL Laponite [71]. A study by Veernala et al. claimed an IC50 of 2.2 mg/mL Laponite [87]. Becher et al. showed IC50 values of approximately 0.5–1.5 mg/mL Laponite when incubated with HeLa or MCF-7 cells [88]. Li et al. reported Laponite concentrations of up to 10% *w*/*v* (100 mg/mL) showed insignificant changes in the cellular viability of pre-osteoblastic MC3T3-E1 cells [89]. Therefore, the range of IC50 values of Laponite varies rather considerably. The concentration of Laponite in the nanocomposite hydrogel delivery devices discussed in this review varies considerably—from 0.05 to 50 mg/mL. However, encapsulation of Laponite within a hydrogel network may reduce the plasma concentration of Laponite particles as the nanoclays are embedded within the hydrogel structure. Laponite particles have been shown to naturally degrade in ~30 days on average (~20–50 days), so a hydrogel with a residence time of >30 days allows for complete Laponite particles to not escape the hydrogel, thereby preventing adverse cytotoxicity [83]. 

Hemolysis and coagulation induced by clay particles is yet another consideration for the use of these nanocomposite systems in vivo. Luo et al. demonstrated that the incorporation of Laponite into a Dextran-based hydrogel did not significantly alter hemolysis [90]. Li et al. showed that the incorporation of Laponite into a gelatin hydrogel improved antithrombogenicity and hemocompatibility [91]. Wang et al. demonstrated Laponite particles showed <5% hemolysis, which could be improved by sintering Laponite particles at high heat [92]. Hence, hemolysis and coagulation should not be a major concern for Laponite–hydrogel composites.

## 7. Laponite–Polymer Composite Hydrogels

Laponite and other nanoclays have been widely shown to improve the mechanical properties of hydrogels when incorporated into the polymer matrix as nanofillers. The nanoclay particles increase the excluded volume within hydrogels, leading to higher stiffness and toughness [93,94]. This toughness can prevent undesired mechanical degradation of hydrogels during or after implantation [95,96]. Furthermore, the addition of Laponite can improve the shear-thinning behavior of hydrogels, which is highly desirable for injectable hydrogel devices where encapsulated drugs and/or cell activity must be preserved during the high shear stress experienced in the injection process [8]. 

Polymers and Laponite particles may interact with each other in several ways [8]. First, nanoclay particles may persist in their stacked, tactoid structure and be effectively phase-separated from the surrounding polymer if affinity between the two species is low, rarely leading to improved mechanical properties [97,98]. Second, polymer chains may be partially intercalated within the inter-particle space of the nanoclays in a “swollen” or more disordered manner. The extent of these interactions depends on the affinity between the polymer chains and Laponite faces [99,100]. Third, Laponite particles may be fully exfoliated into individual sheets with random orientation, surrounded by polymer chains. It is with this third case that hydrogel mechanical properties may be most improved following the addition of Laponite [98,100,101]. The nature of the Laponite particle dispersion within the hydrogel matrix can lead to changes in encapsulation efficiency and drug release kinetics and is, therefore, an important parameter that must be considered when deciding what fabrication method would be used for nanocomposite hydrogels. For example, when Laponite particles were dispersed and interacted with a model small molecule prior to encapsulation within a polyethylene glycol (PEG)-based hydrogel, release was significantly slower than when Laponite particles were embedded within the hydrogel and then subsequently exposed to the small molecule [102]. 

Laponite particles may also be used as physical or chemical crosslinkers to form hydrogel structures. The charged nature of Laponite lends itself to weak interactions between its negatively charged faces or positively charged edges and ionizable moieties of polymers through van der Waals forces or hydrogen bonding [103]. Laponite particles may also be chemically modified to form covalent linkages between polymers to act as a primary or secondary crosslinker [104]. A study by Batista et al. revealed Laponite could enhance photopolymerization conversion for a UV-crosslinked polystyrenesulfonate nanocomposite hydrogel [105]. Modulation of Laponite concentration can also be used to tune the Laponite–polymer interactions, allowing for control of nanocomposite hydrogel mechanical properties [106]. Thus, Laponite–hydrogel composites may exist in a variety of structures and crosslinking mechanisms [105]. 

In addition to forming nanocomposite hydrogel structures with polymers, Laponite particles may self-aggregate into a weak gel above a critical concentration (~20 mg/mL) [107]. Edge-face particle interactions lead to the self-assembly of a “house-of-cards” structure that forms the gel structure [108,109]. These hydrogels are typically much softer than their polymer counterparts and are heavily dependent on local salinity and pH, which can affect their stability [55,110]. Wang et al. demonstrated the loading of the chemotherapeutic doxorubicin into the interlayer space of Laponite particle gels, demonstrating the utility of these nanoclay-only gels for delivery applications [111]. Becher et al. developed Laponite nanogels using inverse mini-emulsion to successfully deliver therapeutics into cancer cells, showing the potential benefits of a Laponite-only nanogel for cellular targeting [88]. 

## 8. Laponite–Hydrogel Nanocomposites for Delivery of Small Molecules

Laponite particles have been widely used in recent years as delivery vehicles for a variety of small molecules and macromolecules due to the benefits of adsorption and/or intercalation on release kinetics. Such examples can be found in reviews by Davis et al. and Kiaee et al. [7,112]. This review specifically focuses on hydrogels containing Laponite. 

Many examples in the literature describe Laponite–hydrogel composites used for the delivery of small molecules (<1000 Da) (Table 2). The large surface area and charge characteristics of Laponite lend to its effectiveness as a delivery vehicle for small molecules. Surface adsorption onto either the faces or edges of Laponite particles provides a facile way to load drugs and therapeutics into hydrogels for subsequent sustained release [113]. Small molecules may also be intercalated into the interlayer space between Laponite particles via cationic exchange [70,114]. Such intercalation is easily determined as the interlayer spacing increases, as observed via X-ray diffraction [102,115]. Following adsorption and/or intercalation of small molecules onto or into Laponite particles, they may be reversibly released due to local salinity, pH, temperature, or Laponite degradation [116]. For example, increased local salinity can cause a cationic exchange of sodium or calcium ions with the intercalated small molecules, thereby leading to release [117]. Charge shielding may also occur at high salt concentrations, leading to an electrostatic double layer that prevents small molecule adsorption [118]. 

A variety of clinically relevant therapeutics have been loaded into Laponite–polymer composite hydrogels. Goncalves et al. described an alginate–Laponite composite hydrogel for the delivery of doxorubicin (Figure 3) [119,120]. Release of the small molecule was significantly slower from nanocomposite hydrogels compared to alginate-only samples, and the release kinetics were contingent on environmental pH. Cancer cells showed reduced viability in the presence of the doxorubicin-loaded nanocomposite hydrogel compared to a bolus drug dose, which was attributed to Laponite particles serving as nanocarriers across the cellular membrane. In another study, β-Lapachone was released from a poly(propylene oxide)-poly(ethylene oxide) (PPO-PEO) block copolymer hydrogel device [121]. The chemotherapeutics’ solubility was increased 40-fold in the presence of Laponite and the delivery device demonstrated cytotoxicity towards cancer cells. These studies, among many others, demonstrate the significantly slowed release of small molecules when interacted with Laponite within composite hydrogels. 

Small-molecule net charge plays a key role in governing release kinetics due to the surface area and charge characteristics of Laponite. Net negative guest molecules are primarily limited to interaction with the edge of Laponite particles, which represents a much smaller surface area than the negatively charged faces of Laponite. Therefore, positively charged molecules (higher pKa) are expected to interact more substantially with Laponite particles. Such trends have been observed in the literature, further corroborating the electrostatic nature of Laponite–small molecule interactions [88,102,117,126]. 

The use of the Laponite–polymer composite system allows for localized delivery and the opportunity for targeted delivery, which is more difficult to achieve with Laponite particles only. Depending on the geometry and size of the bulk hydrogel, Laponite nanocomposite hydrogels may remain at the desired location longer than when individualized [127], since Laponite particles may be rapidly cleared due to their small size. For example, Jiang et al. developed a hyaluronic acid-based device that allowed for specific targeting of CD44-positive cells, improving the efficacy of doxorubicin delivery [123]. 

In addition to the response of Laponite–small molecule interactions to local salinity and pH, encapsulation of Laponite particles allows for the complete nanocomposite hydrogel to exhibit stimuli-responsive properties. A dextran-based hydrogel containing Laponite exhibited stimuli-responsiveness to near-infrared stimulation, providing a tuned release of ciprofloxacin [90]. Gharaie et al. developed a pH-responsive gelatin-based hydrogel that incorporated Laponite, providing varying release kinetics of rhodamine B, a model small molecule [128]. Such stimuli-responsive delivery devices allow for another lever by which release can be modulated in addition to Laponite–drug interactions.

## 9. Laponite Composite Hydrogels for Delivery of Macromolecules

While 2D silicate nanoclays have been widely used for the adsorption and delivery of small molecules, much less research has been devoted to macromolecules such as proteins and nucleic acids. The larger size of these macromolecules may lead to varying interactions with nanoclays compared to small molecules [129]. Proteins have been shown to form relatively large complexes with Laponite, with the size of these complexes being contingent on protein charge and nanoclay concentration [39,69]. Positively charged proteins form larger complexes with Laponite than do negatively charged proteins, which can be attributed to the positively charged proteins having a larger nanoclay surface area on which to bind than the negatively charged proteins, which are primarily limited to adsorption on the relatively small nanoclay edges. However, macromolecules such as proteins may exhibit surface-patch binding onto Laponite particles, in which proteins “bind across their pH” [69]. For example, a positively charged region on an overall negatively charged protein may interact with the Laponite particle face. Therefore, protein and other macromolecules can form complex interactions with Laponite particles and are not limited to face-only or edge-only interactions. 

Similarly, Kim et al. adsorbed albumin and lysozyme to Laponite in a hyaluronic acid hydrogel that was physically crosslinked by modified Laponite particles [104]. In this system, Laponite served a dual purpose: Laponite edges served as physical crosslinking sites to interact with the polymer to form a hydrogel; Laponite faces were uninvolved with the polymer to remain available for adsorption of proteins. As expected, negatively charged albumin was released faster than positively charged lysozyme. Importantly, the release was shown in a mouse model, demonstrating in vivo release kinetics and retention of protein bioactivity to induce osteogenic effects. In another example, Koshy et al. demonstrated protein adsorption and sustained release of five clinically relevant proteins with varying sizes and charges from a Laponite-containing alginate click hydrogel [130]. Proteins were incubated with Laponite and then encapsulated within the hydrogel. Burst release was mitigated for all proteins and release kinetics were demonstrated to be contingent on Laponite concentration, allowing for tunable release times. A study by Li et al. utilized an alginate/Laponite nanocomposite hydrogel device in which Laponite particles were complexed with insulin-like growth factor-1 mimetic protein (ILGF-1) and subsequently entrapped within an alginate hydrogel [131]. ILGF-1 release kinetics were dependent on Laponite concentration and showed release up to 4 weeks in a rat model. While these studies demonstrated successful electrostatic interactions between Laponite and protein, as well as slowed release kinetics, there are few studies elaborating the stability and structure of Laponite–protein intermediaries.

In a paper by Stealey et al., Laponite was incubated with three model proteins of varying size and charge (Figure 4) to investigate Laponite–protein interactions. Laponite–protein complex size increased with increasing Laponite concentration due to the increase in surface area available for adsorption. Importantly, the buffer in which Laponite was dispersed played a key role in determining Laponite–protein complex size, with high osmotic buffers triethanolamine and phosphate buffered saline showing little particle exfoliation or Laponite–protein complex formation. Conversely, deionized water allowed for facile exfoliation and interaction with proteins, thereby resulting in large Laponite–protein complexes. 

The formation of these nanoclay–protein complexes also showed a significantly slowed release of three model proteins from PEG–nanoclay composite hydrogels in vitro. Positively charged ribonuclease A (RNase) and lysozyme (Lys) were released up to 23 times slower following complexation with Laponite, compared to PEG-only hydrogels. Negatively charged bovine serum albumin (BSA) was also released significantly slower in the PEG–nanosilicate hydrogels, though this effect was less profound than for the positively charged proteins. This can once again be attributed to the formation of smaller Laponite–protein complexes for negatively charged proteins. While this research gave more insight into the formation of Laponite–protein complexes, the stoichiometry, structure, reversibility, and stability of the complexes remain unresolved. 

Because of the importance of protein secondary and tertiary structure on bioactivity, protein structure must be preserved following interaction and release from Laponite. Cross et al. demonstrated the binding of human bone morphogenetic protein 2 (rhBMP2) and transforming growth factor-β3 (TGF-β3) with Laponite particles [132]. Proteins exhibited sustained release following adsorption onto Laponite particles in the absence of a polymeric hydrogel. Importantly, osteogenic effects were observed in a 2D cell culture model following the release of proteins, demonstrating that released proteins remained bioactive following interaction with Laponite. In another example of proteins preserving their bioactivity, gelatin methacrylate–Laponite nanocomposite hydrogels were fabricated by Waters et al. that incorporated human mesenchymal stem cell-derived growth factors [133]. Sustained release of vascular endothelial growth factor (VEGF) and fibroblast growth factor 2 (FGF2) was demonstrated in vitro. The secretome-loaded nanocomposite hydrogel demonstrated the potential to enhance angiogenesis and cardio-protection. In another study that demonstrated the utility of Laponite composite hydrogels, Liu et al. developed an alginate/gelatin/Laponite nanocomposite hydrogel system that was shown to be noncytotoxic and could successfully deliver bone marrow mesenchymal stem cells in a critical-size rat bone calvarial defect [134]. The degradation products of Laponite were shown to enhance the osteogenic potential compared to hydrogels without Laponite. The authors suggested that this nanocomposite hydrogel system could also be loaded with drugs to further enhance bone regeneration. Together, these studies show proteins retain at least a portion of their bioactivity following release from Laponite–hydrogel composites and can achieve desired physiological outcomes.

In a unique hydrogel fabrication method, Dawson et al. fabricated self-assembling Laponite hydrogels for protein delivery [135]. Laponite was added dropwise above a critical Laponite concentration to varying concentrations of NaCl to form microcapsules. These Laponite microcapsules showed sustained release of albumin and lysozyme compared to bolus dose and control alginate hydrogels. VEGF was then loaded into the Laponite microcapsules and demonstrated enhanced angiogenesis in 2D cell culture, thereby demonstrating retention of protein bioactivity following interaction with Laponite particles. A collagen scaffold was then used to subencapsulate the Laponite microcapsules to release VEGF and BMP2 in a rat model, which showed enhanced angiogenesis in vivo. This group further demonstrated these Laponite gel capsules could successfully localize the release of BMP2 to achieve ectopic bone formation in a rat model due to the release and preservation of protein bioactivity [136]. Therefore, Laponite may be used as a gel-forming agent in the absence of polymers, while still achieving sustained release and retained bioactivity. 

The surface of Laponite particles may also be modified via interactions with proteins to promote more specific or effective binding of another target protein. Wang et al. designed a blended hydrogel system with heparin and Laponite to deliver fibroblast growth factor 4 (FGF4) for the treatment of spinal cord injury [137]. Heparin was first reacted with FGF4 to form a heparin protein complex, which was then adsorbed onto Laponite particles. The system exhibited sustained release of over 35 days and enhanced the recovery process in a rat model. 

Together, these studies reveal the utility of Laponite–hydrogel composites for the delivery of macromolecules where release kinetics are slowed and molecular bioactivity is maintained. However, more research must be conducted to determine the stoichiometry, geometry, and stability of the formed Laponite–protein complexes. Additionally, an understanding of protein activity while adsorbed to Laponite is also desired to determine whether bioactivity is retained throughout the adsorption and subsequent release, or if a temporary (or permanent) unfolding of protein occurs during interaction with Laponite. Without knowledge of the processes that govern the Laponite–protein interactions, the use and tunability of Laponite nanocomposite hydrogels as drug delivery devices may not be fully realized. 

## 10. Potential Challenges and Drawbacks of Laponite Composite Hydrogels

While the incorporation of Laponite within hydrogels offers a plethora of benefits, some challenges may persist that may delay or hinder their clinical use. Because of the prolonged release profiles of drugs afforded by electrostatic interaction with Laponite, release may be too slow for some applications. For example, the use of a Laponite hydrogel composite may not be appropriate for applications where a relatively high drug plasma concentration is needed for only a short time. Laponite may delay delivery of the drug and result in ineffective dosing. Furthermore, Laponite–drug interactions may last longer than the typical degradation time of the hydrogel and or Laponite particle itself. This may result in a pseudo burst release or at least steep increase in release kinetics due to escape and/or degradation of Laponite particles, thereby releasing adsorbed cargo. However, such a secondary burst release does not appear in the literature to our knowledge. Consideration must also be given to optimization of Laponite concentration within hydrogels. While increasing Laponite concentration may lead to slower release kinetics, too high of a Laponite concentration may adversely affect hydrogel mechanical properties due to hinderance of desired crosslinking. When used for protein delivery, Laponite composite hydrogels must not irreversibly denature or unfold proteins, rendering them inactive. Examples in the literature seemingly indicate that released proteins retain their bioactivity, but further characterization and understanding of the Laponite–protein complex structure is necessary. Another potential complication would be the behavior of Laponite–drug complexes in physiological fluids, which are rich with a variety of small molecules and proteins. Understanding how these other molecules affect Laponite–drug interactions and stability is paramount to achieving controllable release profiles [138]. 

## 11. Conclusions and Future Directions

Laponite–hydrogel composites offer great potential for use as devices for the delivery of both small molecules and macromolecules because of the unique benefits of the nanocomposite system that cannot be achieved with just a hydrogel or just Laponite. Drugs may be adsorbed or intercalated onto or into Laponite particles, significantly reducing burst release and lengthening the duration of sustained release. For both small molecules and proteins, guest molecule net charge plays a key role in determining release kinetics due to the unique charge and surface area characteristics of Laponite. Release kinetics may also be governed by Laponite concentration and environmental pH and/or salinity. Importantly, release drugs have been shown to retain their bioactivity to achieve desired in vivo responses. In addition to the benefits of controlling release, the incorporation of Laponite particles can positively affect hydrogel mechanical and physical properties, making them even more suitable as injectable delivery devices. Encapsulating Laponite within hydrogels also allows for tunable, localized release, with the nanocomposite hydrogel serving as a depot for drug release.

Going forward, more in-depth research must be conducted on the nature of Laponite–macromolecule complexes to fully harness the power of these nanocomposite hydrogel delivery devices. An understanding of how macromolecules of varying sizes and charges interact with Laponite will allow for enhanced tunability of release profiles for specific applications. Currently, a significant gap in the literature exists in determining the stability, reversibility, and stoichiometry of Laponite–protein complexes. Knowledge of potential protein unfolding or denaturation when adsorbed to Laponite is crucial to ensure effective protein delivery. Furthermore, the delivery of macromolecules other than proteins should be explored, such as that of nucleic acids and immunoglobulins, which each represent their challenges. 

In the future, Laponite-based nanocomposite hydrogels are anticipated to have widespread applications in tissue engineering and regenerative medicine. They may be utilized in personalized medicine, enabling patient-specific hydrogels, as well as in the development of smart hydrogels with stimuli-responsive drug release. Integration with bioelectronics and sensors could facilitate real-time monitoring, while bioprinting techniques could allow for the creation of complex tissue constructs. Combination therapies, bioactive coatings for implants, and integration with AI for predictive modeling are also potential advancements. Additionally, these nanocomposite hydrogels show promise in additive biomanufacturing, particularly in extrusion-based bioprinting, where the inclusion of Laponite enables shear-thinning behavior. Therapeutics can be incorporated into printed structures to guide cellular functions, allowing for the creation of heterogeneous tissue architectures. This approach has the potential to revolutionize drug testing and accelerate the clinical translation of therapeutics.

As more hydrogel–Laponite composite delivery devices emerge, we will obtain a better knowledge of how the devices perform in vivo and interact with cells and blood in physiological environments. While Laponite-containing devices have shown great promise so far, we must be able to translate the beneficial release kinetics in a physiological environment where numerous other molecules will compete for interactions with Laponite particles. 

## Figures and Tables

**Figure 1 pharmaceuticals-16-00821-f001:**
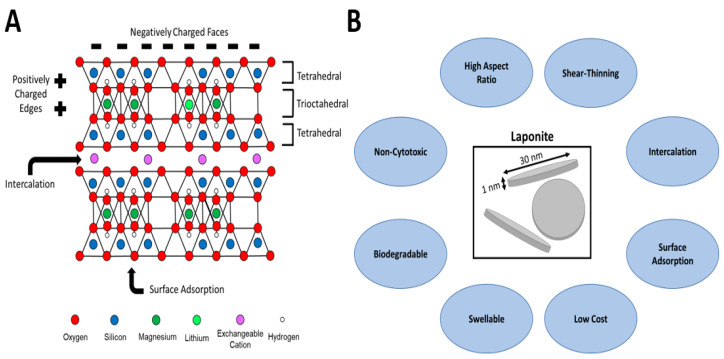
(**A**). Structure of Laponite with 2:1 tetrahedral:trioctahedral layering, allowing for intercalation and adsorption of drug molecules. (**B**). Properties of Laponite particles making them beneficial for use in drug delivery applications.

**Figure 2 pharmaceuticals-16-00821-f002:**
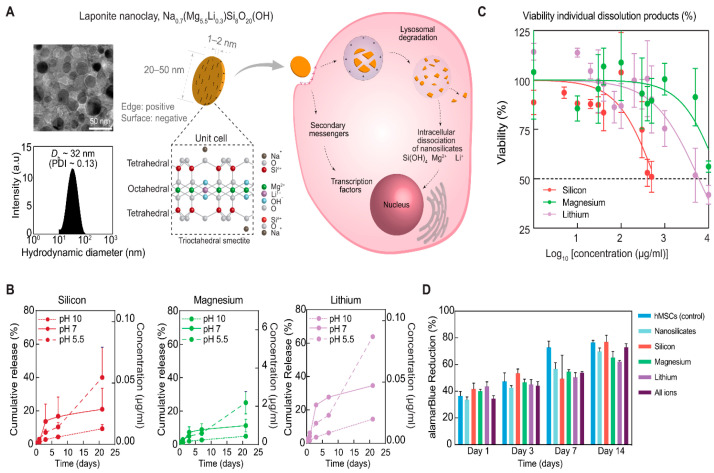
Structure, physiological stability, and cellular compatibility of Laponite. (**A**) Laponite (nanosilicates, nSi) are plate-like poly-ions composed of simple or complex salts of silicic acids with a heterogeneous charge distribution and patchy interactions. Transmission electron microscopy (TEM) images show the size of Laponite to be between 20 and 50 nm in diameter. Dynamic light scattering (DLS) shows the hydrodynamic diameter (Dh) of Laponite to be ~32 nm in aqueous conditions, with a polydispersity index (PDI) of ~0.13. The schematic shows the potential interactions of Laponite with cells. Laponite dissociates into individual ions once introduced to a physiological microenvironment (pH < 9). a.u., arbitrary units. (**B**) The dissolution of Laponite was monitored using inductively coupled plasma mass spectrometry (ICP-MS) at different pH to mimic the extracellular (pH~7.4) and intracellular (pH~5.5) microenvironments. Laponite is expected to be stable at pH~10, and thus, pH 10 was used as control. (**C**) The effect of Laponite and its ionic dissolution products (silicon, magnesium, and lithium) on cellular viability was evaluated using an MTT assay. Three technical replicates were used for each condition. Half-maximal inhibitory concentration (IC50) is labeled at 50% viability. Concentrations of released ions from Laponite fall well below the IC50 value. (**D**) Long-term cellular viability after treatment with nanoparticles and its ionic dissolution products was assessed using an alamarBlue assay to detect metabolically active cells (n = 3). Adapted with permission from Ref. [83]. 2022. Science.

**Figure 3 pharmaceuticals-16-00821-f003:**
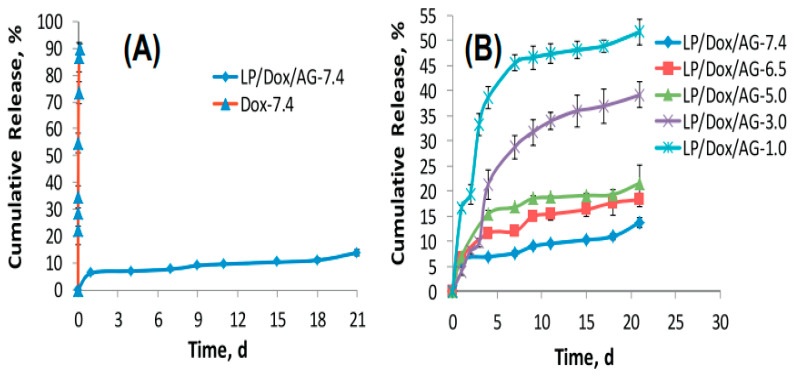
Release of free Doxorubicin (DOX) and DOX from Alginate (AG)/Laponite (LP) nanocomposite hydrogels at pH 7.4 (**A**) and at different pH values in PBS (**B**). Reprinted with permission from Ref. [119]. 2014, Elsevier.

**Figure 4 pharmaceuticals-16-00821-f004:**
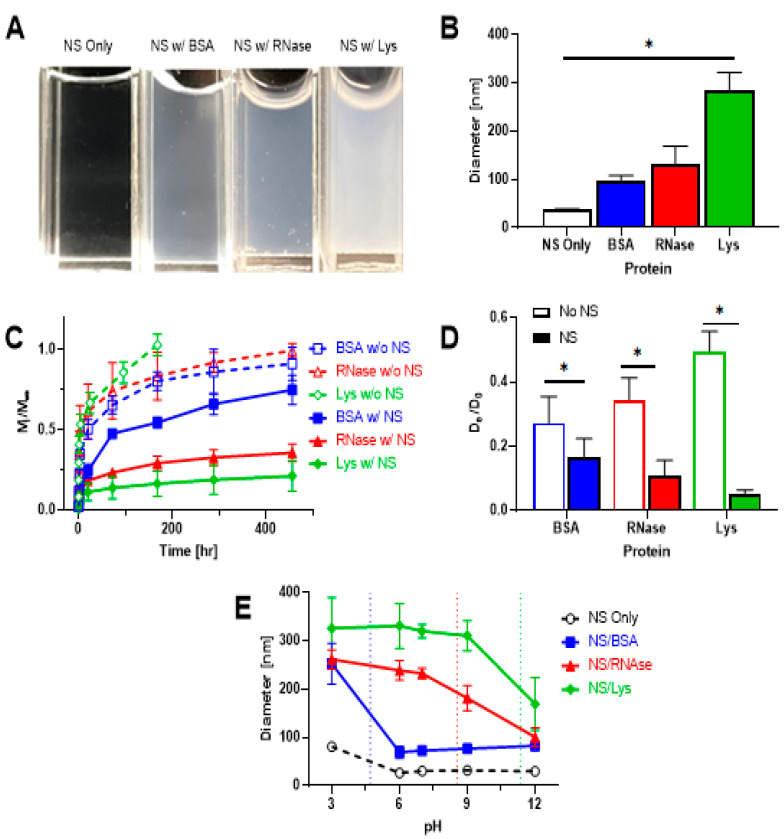
(**A**). Visual observation of complexation of nanosilicates (NS; 10 mg/mL) with Lys, BSA, and RNase (2 mg/mL). (**B**). Diameter of NS only (no protein) and NS–protein complexes measured via dynamic light scattering. NS concentration was 1 mg/mL and protein concentration was 1 mg/mL. * Indicates significant difference (N = 6, *p* < 0.05). (**C**). Release profiles of BSA, RNase, and Lys from PEG-only (dashed lines) and NS–PEG (10 mg/mL NS, solid lines) hydrogels. (**D**). Normalized diffusivity of proteins in PEG-only (No NS) compared to NS–PEG hydrogels. * indicates statistically significant difference (N = 6, *p* < 0.05). (**E**). Diameter of NS only (no protein) and NS–protein complexes as a function of pH. NS (100 μg/mL) and protein (50 μg/mL) were incubated for 30 min before measurements. Vertical dashed lines represent isoelectric points of BSA (blue; pI = 4.7), RNase (red; pI = 8.54), and Lys (green; pI = 11.35). Republished with permission from Ref. [39]. 2021. American Chemical Society.

**Table 1 pharmaceuticals-16-00821-t001:** Chemical formula and diameter of smectite nanoclays [53,54,55,56,57,58,59].

Silicate Nanoclay	Chemical Formula	Cationic Exchange Capacity [meq/g]
Montmorillonite	(Na,Ca)_0.33_(Al,Mg)_2_(Si_4_O_10_)(OH)_2_ · nH_2_O	1.2
Hectorite	Na_0.3_(Mg,Li)_3_(Si_4_O_10_)(F,OH)_2_	0.6
Saponite	Ca_0.25_(Mg,Fe)_3_((Si,Al)_4_O_10_)(OH)_2_ · nH_2_O	0.1
Nontronite	Na_0.3_Fe_2_((Si,Al)_4_O_10_)(OH)_2_ · nH_2_O	0.5
Beidellite	(Na,Ca_0.5_)_0.3_Al_2_((Si,Al)_4_O_10_)(OH)_2_ · nH_2_O	0.7
Laponite	Na_0.7_Si_8_Mg_5.5_Li_0.3_O_20_(OH)_4_	0.5

**Table 2 pharmaceuticals-16-00821-t002:** Examples of Laponite–polymer composite hydrogels for the delivery of small molecules (<1000 Da), including the application of the encapsulated molecule, the polymer used for hydrogel fabrication, the small molecule delivered, and whether the study included in vivo experiments.

Application	Polymer	Small Molecule Delivered	In Vivo Studies	References
Anti-Cancer	Alginate	Doxorubicin	Yes	[119,120]
	PEG	Acridine Orange, Doxorubicin, Alexa 546	No	[102,117]
	PPO-PEO	β-Lapachone	No	[121]
	Hyaluronic Acid	Methotrexate	Yes	[122]
	None	Cisplatin, 4-fluorouracil, cyclophosphamide	Yes	[88]
	Hyaluronic Acid	Doxorubicin	No	[123]
Anti-Bacterial	Chitosan	Ofloxacin	No	[124]
	Dextran	Ciprofloxacin	No	[90]
Anti-Inflammatory	Gellan Gum	Theophylline, vitamin B12	No	[125]

## Data Availability

No new data were created or analyzed in this study. Data sharing is not applicable to this article.

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
