# Peer review of "Laponite-Based Nanocomposite Hydrogels for Drug Delivery Applications"

_pharmaceuticals, 2023, doi:10.3390/ph16060821_

Round 1
Reviewer 1 Report
This article concentrates on Laponite, a nanosilicate with a 30-nm diameter and 1-nm thickness. The advantages of incorporating Laponite into hydrogels are discussed, along with examples of Laponite-hydrogel composites that are presently under investigation for their ability to delay the release of small molecules and macromolecules such as proteins. Future research will further characterize the interaction between nanosilicates, hydrogel polymer, and encapsulated therapeutics, as well as the effect of each of these components on release kinetics and mechanical properties.
Introduction
Provide the references for
"However, a notable challenge with hydrogels is their susceptibility to the initial burst release of loaded therapeutics"
"To address this issue, several techniques have been employed to minimize this burst release and achieve sustained drug release kinetics"
"Extensive research is being conducted on a wide range of nanoparticles, including polymeric, carbon-based, metal, metal oxides, and ceramic nanoparticles, to reinforce hydro gel networks for drug delivery applications"
"The combination of high surface area and charge also results in the sustained release of loaded therapeutics"
How similar headings are given for two sections??
9. Laponite Composite Hydrogels for Delivery of Macromolecules
10. Laponite Composite Hydrogels for Delivery of Macromolecules
In the start it was planned that "In this review, we critically evaluate the use of Laponite-based nanocomposite hydrogels for drug delivery applications"
but in whole manuscript there is no critical comparison or discussion it all just compiling of data. Its strongly suggested to revise the all section by adding critical comparison with all other related used materials.
There should be a proper conclusion and direction for future work.
Check for mistakes as same headings are repeated,
Reviewer 2 Report
Review of the manuscript which has been submitted to Pharmaceuticals
Manuscript- pharmaceuticals-2410274
In the current context of the study topic, the manuscript entitled “Laponite-based Nanocomposite Hydrogels for Drug Delivery Applications” is well chosen and very interesting.
Below I have some observations and questions to improve the quality of the work.
· The Conclusion and Author Contributions sections are missing.
· The Sections 9 and 10 have the same name!
Reviewer 3 Report
The manuscript provided a comprehensive review regarding the development of laponite incorporated hydrogel for drug delivery. The review is generally well written and easy to follow. I only have some small questions below:
1, It will be great if the authors can give some introduction regarding how the Laponite is synthesized.
2, Is there any drawback for the Laponite based delivery system? Based on the authors' summary, Laponite seems a very promising delivery system, but why number of the drug delivery studies appears to be limited?
Round 2
Reviewer 1 Report
Authors have done required corrections.
Acceptable.